# Quantum Spacetime Geometrization: QED at High Curvature and Direct Formation of Supermassive Black Holes from the Big Bang

**Piero Chiarelli** [1,2]

1   National Research Council of Italy, San Cataldo, Moruzzi 1, 56124 Pisa, Italy; pchiare@ifc.cnr.it;
    Tel.: +39-050-315-2359; Fax: +39-050-315-2166
2   Interdepartmental Center "E. Piaggio", Faculty of Engineering, University of Pisa, Diotisalvi 2, 56122 Pisa, Italy

**Abstract:** In this work, the author employs the quantum hydrodynamic formalism to achieve the geometrization of spacetime for describing the gravitational interaction within the framework of quantum theory. This approach allows for the development of an equation of gravity that is mathematically connected to the fermion and boson fields. This achievement is accomplished by incorporating two fundamental principles: covariance of the quantum field equations and the principle of least action. By considering these principles, a theory is established that enables the calculation of gravitational corrections to quantum electrodynamics and, potentially, to the standard model of particle physics as well. The theory also provides an explanation for two phenomena: the existence of a cosmological pressure density similar to quintessence, which is compatible with the small value of the observed cosmological constant, and the breaking of matter–antimatter symmetry at high energies, offering insights into why there is an imbalance between the two in the early universe. In the cosmological modeling of the theory, there exists a proposal to account for the formation of supermassive black holes that are accompanied by their own surrounding galaxies, without relying on the process of mass accretion. The model, in accordance with recent observations conducted by the James Webb Space Telescope, supports the notion that galactic configurations were established relatively early in the history of the universe, shortly after the occurrence of the Big Bang.

**Keywords:** quantum gravity; quantum electrodynamics; quantum cosmology

## 1. Introduction

Physics faces a major challenge in unifying general relativity and quantum theory, which are fundamentally different in their approaches. General relativity explains gravity as the curvature of spacetime due to matter and energy, while quantum theory describes the behavior of matter and energy at very small scales. However, attempts to merge these two theories have been unsuccessful so far. There have been various proposals for reconciling general relativity and quantum theory, such as string theory [1–3], loop quantum gravity [4–6], and causal dynamical triangulation [7–9], but none have yet been confirmed by experimental evidence. The search for a unified theory of physics remains an active area of research in theoretical physics.

Despite its progress and successes, loop quantum gravity (LQG) also has some limitations and shortcomings, as follows:

1.  Lack of a complete formulation. Currently, LQG does not have a complete and well-defined formulation. There is no unified and coherent description that covers all spatial and temporal scales. This limits the theory's ability to provide definitive answers to important questions about the nature of quantum gravity.
2.  Problems with Newtonian gravity and general relativity. LQG fails to directly recover Newtonian gravity or Einstein's general relativity in the classical limit. Although it is

desirable for a theory of quantum gravity to include general relativity as a limiting case, it has not been fully demonstrated how this occurs in LQG.

3.  Ambiguity in the quantization of continuous geometries. In LQG, the geometry of space is quantized, but the approach used to carry out this quantization can lead to ambiguities and uncertainties. This can affect the precision of theoretical predictions and the consistency of the theory itself.

4.  Challenges in comparison with experimental observations. LQG is still in a development phase and has not yet produced specific predictions that can be directly compared with experimental observations. This makes it difficult to verify the validity or invalidity of the theory through current experimental methods.

String theory attempts to unify all the fundamental forces and particles of nature by describing them as vibrating strings, but it faces several challenges and shortcomings:

1.  Lack of experimental confirmation. One of the significant challenges facing string theory is the lack of direct experimental evidence. Due to the extremely high energies required to observe string-like behavior, it has been challenging to test the theory through experiments. As a result, it has not yet made specific predictions that can be validated or refuted by experimental data.

2.  Landscape problem and vacuum solutions. String theory predicts the existence of a vast landscape of possible vacuum solutions, each representing a different physical universe. However, identifying the specific vacuum solution that corresponds to our observed universe remains an open question. This raises concerns about the uniqueness and predictability of the theory.

3.  Fine-tuning and naturalness. String theory often requires fine-tuning of parameters to match the observed values of fundamental constants in our universe. This fine-tuning can be seen as a drawback, as it raises questions about the naturalness and predictability of the theory. Additionally, the precise mechanisms that select the particular vacuum state and fix the parameters are not yet fully understood.

4.  Emergence of spacetime and gravity. String theory suggests that spacetime and gravity emerge from underlying microscopic degrees of freedom. However, the exact mechanism of this emergence is still not well understood. The transition from a fundamental string theory to a macroscopic, four-dimensional spacetime with gravity is a challenging problem that requires further development.

Quantum field theory in curved spacetime of general relativity has been successful in various contexts, such as the semiclassical approximation, but it also faces certain challenges and limitations:

1.  Lack of a complete and consistent formulation. Quantum field theory in curved spacetime does not have a unique and fully consistent formulation that covers all possible curved spacetimes. The theory encounters conceptual and technical difficulties when applied to highly curved or dynamically changing spacetimes, such as those near black holes or during the early universe. This makes it challenging to obtain precise and reliable predictions in such regimes.

2.  Renormalization issues. Renormalization, a procedure used to remove infinities from calculations in quantum field theory, becomes more complicated in curved spacetime. The presence of curved backgrounds introduces additional divergences and ambiguities, making the renormalization process more involved and less well-understood. This can limit the predictive power and precision of the theory.

3.  Backreaction problem. In quantum field theory in curved spacetime, the backreaction of the quantum fields on the geometry of spacetime is often neglected or treated in a simplified manner. This neglecting of the backreaction can lead to inconsistencies and limitations in the theory (in high gravity, at the Big Bang, and in the early universe). Properly accounting for the backreaction in a self-consistent manner remains a challenge.

4. Ultraviolet divergences and quantum gravity. The presence of ultraviolet divergences (infinities arising at very high energies) in quantum field theory can be incompatible with the quantization of gravity. This indicates that a more fundamental theory, such as a theory of quantum gravity, might be necessary to consistently describe physics in highly curved spacetimes.

5. Limited applicability beyond perturbative regimes. Quantum field theory in curved spacetime is often formulated and applied within the framework of perturbation theory, which assumes that the spacetime curvature is small compared to a flat background. This limits its applicability to situations where the gravitational fields are weak and the deviations from flat spacetime are small. Exploring quantum field theory in strongly curved or highly dynamical spacetimes remains a challenge.

On the other hand, there are several problems of general relativity with respect to the quantum nature of the matter, which is believed to be inadequate in fully explaining some observed phenomena in the universe. One of its shortcomings comes from the fact that in general relativity, the energy–momentum tensor density, $T_{\mu\nu}$ for massive bodies depends on the point-dependent mass density and is limited to the reductive classical expression, while the general form: $T_{\mu\nu} = (p + \varepsilon)u_\mu u_\nu + pg_{\mu\nu}$, that can host quantum effects remains undefined. The semiempirical inclusion of the cosmological constant within general relativity exemplifies the necessity to address this deficiency in the theory. However, akin to any semiempirical approach, the parameters involved are not derived from fundamental physics but rather considered as constants to be measured. Consequently, this fundamentally renders it impossible to rationalize the remarkably low measured cosmological constant in comparison to the vacuum's zero-point energy.

Moreover, in classical general relativity, the energy–momentum tensor that describes the distribution of energy and momentum in spacetime is a symmetric tensor with the off-diagonal terms are zero. This symmetry is a consequence of assuming a smooth and continuous geometry of the spacetime. The non-commutativity of spatial rotations in three or more dimensions serves as a prevalent example of non-commutative operations. Non-commutativity stands as a key mathematical concept that articulates uncertainty in quantum mechanics, manifesting in any pair of conjugate variables, such as position and momentum. In the presence of a magnetic field, even momenta no longer mutually commute.

Building on this foundation, when exploring the impacts of quantum mechanics and the potential non-commutative properties of spacetime, some theories propose that the geometry of spacetime could become non-commutative.

Moreover, one can just imagine that position measurements might fail to commute and describe this using non-commutativity of the coordinates. Building upon this concept, a straightforward modification to quantum fields, involving coordinates satisfying non-commutation relations, allows the definition of a broad class of non-commutative field theories. This idea has spurred extensive research in recent decades [10].

The resulting non-commutativity is evident in the presence of off-diagonal terms in the energy–momentum tensor.

In a recent study [11], the author showcased that by postulating the covariance of quantum mechanical field equations and employing their quantum hydrodynamic portrayal, it becomes feasible to delineate the spacetime geometry via a gravity equation that encompasses quantum mechanics. This accomplishment is realized by employing a generalized least-action principle, yielding a set of equations that characterizes the evolution of quantum–gravitational interactions. The resulting system intertwines the gravity equation with the quantum equation of bosonic or fermionic fields. The theory, based on the covariance of quantum mechanics, naturally introduces non-commutative terms into the energy–momentum tensor, generating a self-defined (quintessence-like) cosmological energy pressure density that is not reliant on a field of elusive physical origin, but rather emerges from the quantum nature of spacetime. This cosmological energy pressure density exhibits a significant magnitude in compact objects, such as black holes and supermassive

black holes, while vanishing in macroscopic, weak gravitational matter. As a result, this naturally leads to an average value throughout the universe that aligns with the observed extremely low value.

Additionally, the theory incorporates an analytical relationship between gravity and the fields, capturing the field backreaction on gravity and enabling the description of field evolution across various physical scales, including high-gravity regimes. The model possesses several noteworthy advantages.

It resolves the issue of point singularities present in general relativity [12] through the repulsive force exerted by the quantum potential, which embodies quantum physics within the framework of quantum hydrodynamic representation.

It also explains repulsive Newtonian gravity at large distances, as generated by the cosmological pressure density tensor in the presence of a background of stochastic gravitational noise (dark energy).

The theoretical investigation put forth in this paper holds the promise of advancing quantum electrodynamics (QED) for high-energy fermionic states and exploring gravitational corrections to Minkowskian theory. The theory reveals symmetry-breaking in gravitational fermion–antifermion interactions, as well as the potential discrepancy in the magnetic moments of leptons and antileptons, which could serve as a possible test for the theory.

## 2. The Charged Fermion Field Coupled to the Gravity Equation

In a recent paper [11], the author demonstrated that it is possible to define the geometry of the spacetime induced by fermion and boson fields by utilizing the covariance condition and the least-action principle. This achievement is accomplished by employing the Madelung quantum hydrodynamic description, where the field equation for the complex field: $\Psi = |\Psi|exp[i\frac{S}{\hbar}]$, is transformed into a system of differential equations in terms of the real variables $|\psi|^2$ and $S$ [11]. Namely, applying this procedure for the Dirac equation in curved spacetime:

$$\left(i\hbar\gamma^\mu\left(\partial_\mu - \frac{i}{4}\sigma^{ab}\omega_{ab\mu} + \frac{ie}{\hbar}A_\mu\right) - mc\right)\Psi = 0, \tag{1}$$

where:

$$\sigma^{ab} = \frac{i}{2}[\gamma^a, \gamma^b] \tag{2}$$

$$\omega_{ab\mu} = f_b^\alpha e_{a\beta}\Gamma_{\mu\alpha}^\beta - f_b^\alpha \partial_\mu e_{a\alpha}, \tag{3}$$

where $e_b^\alpha$ and $f_b^\alpha$ are the vielbein and the inverse vielbein, respectively, and $\Gamma_{\mu\alpha}^\beta$ is the Christoffel matrix, which reads:

$$\Gamma_{\mu\alpha}^\beta = \frac{1}{2}g^{\beta\gamma}\left(\partial_\alpha g_{\gamma\mu} + \partial_\mu g_{\gamma\alpha} - \partial_\gamma g_{\mu\alpha}\right), \tag{4}$$

by utilizing the substitution:

$$\psi_\pm = \frac{\psi_1 \pm \psi_2}{\sqrt{2}} = \begin{pmatrix} \psi_{\pm 1} \\ \psi_{\pm 2} \end{pmatrix} = \begin{pmatrix} |\psi_{\pm 1}|exp[i\frac{S_{\pm 1}}{\hbar}] \\ |\psi_{\pm 2}|exp[i\frac{S_{\pm 2}}{\hbar}] \end{pmatrix} = \begin{pmatrix} \sum\limits_{k=0} \alpha_{\pm 1k}exp[-i\frac{S_{\pm 1k}}{\hbar}] + \beta^\dagger_{\pm 1k}exp[i\frac{S_{\pm 1k}}{\hbar}] \\ \sum\limits_{k=0} \alpha_{\pm 2k}exp[-i\frac{S_{\pm 2k}}{\hbar}] + \beta^\dagger_{\pm 2k}exp[i\frac{S_{\pm 2k}}{\hbar}] \end{pmatrix}, \tag{5}$$

where $\begin{pmatrix} \psi_1 \\ \psi_2 \end{pmatrix} = \Psi$, the system of four differential equations in terms of the real variables $|\psi_{\pm 1}|^2$, $|\psi_{\pm 2}|^2$, $S_{\pm 1}$ and $S_{\pm 2}$ [11] follows.

By assuming the covariant derivative for affine and spinor connections:

$$D_\mu = \partial_\mu - \frac{i}{4}\sigma^{ab}\omega_{ab\mu}. \tag{6}$$

Equation (1) more simply reads:

$$\left(i\hbar\gamma^\mu\left(D_\mu + \frac{ie}{\hbar}A_\mu\right) - mc\right)\Psi = 0, \tag{7}$$

which by utilizing Equation (5) leads to:

$$\frac{i\hbar}{mc}\sigma^\mu\left(D_\mu + \frac{ie}{\hbar}A_\mu\right)\psi_+ = \psi_-, \tag{8}$$

$$\frac{i\hbar}{mc}\widetilde{\sigma}^\mu\left(D_\mu + \frac{ie}{\hbar}A_\mu\right)\psi_- = \psi_+, \tag{9}$$

from which the covariant form of the Dirac equation, as a function of the fields $\psi_\pm$, reads:

$$\left(g^{\mu\nu} + \alpha_\pm^{\mu\nu}\right)\left(\partial_\mu - \frac{i}{4}\sigma^{ab}\omega_{ab\mu} + \frac{ie}{\hbar}A_\mu\right)\left(\partial_\nu - \frac{i}{4}\sigma^{ab}\omega_{ab\nu} + \frac{ie}{\hbar}A_\nu\right)\psi_\pm = -\frac{m^2c^2}{\hbar^2}\psi_\pm, \tag{10}$$

which, after some manipulation, leads to [11]:

$$\left(\begin{array}{c}\left(\partial_\mu - \frac{i}{4}\sigma^{ab}\omega_{ab\mu} + \frac{ie}{\hbar}A_\mu\right)\left(g^{\mu\nu}\left(\partial_\nu - \frac{i}{4}\sigma^{ab}\omega_{ab\nu} + \frac{ie}{\hbar}A_\nu\right)\right) \\ +g^{\mu\beta}g^{\nu\alpha}\alpha_{\pm\beta\alpha}\left(\frac{ie}{2\hbar}F_{\mu\nu} - \frac{i}{4}\sigma^{ab}\left(\omega_{ab\mu}\partial_\nu + \partial_\mu\omega_{ab\nu}\right)\right)\end{array}\right)\psi_\pm = -\frac{m^2c^2}{\hbar^2}\psi_\pm. \tag{11}$$

Moreover, by equating the real and imaginary parts, respectively, of Equation (11), the four hydrodynamic quantum equations are obtained. The real part leads to the Hamilton–Jacobi hydrodynamic motion equation in curved spacetime, which reads [11]:

$$\begin{array}{c}\left(\partial_\mu S_\pm - \left(Re\left\{\frac{\hbar}{4}\sigma^{ab}\omega_{ab}\mu\right\} - eA_\mu\right)\right)g^{\mu\nu}\left(\partial_\nu S_\pm - \left(Re\left\{\frac{\hbar}{4}\sigma^{ab}\omega_{ab}\nu\right\} - eA_\nu\right)\right) \\ = \left(m^2c^2 - mV_{qu\pm}\right)\end{array}, \tag{12}$$

where the quantum potential, $V_{qu\pm}$, reads [11]:

$$V_{qu\pm} = -\frac{\hbar^2}{m}\left(\begin{array}{c}\left(\begin{pmatrix}\frac{\partial_\mu(g^{\mu\nu}\partial_\nu|\psi_{\pm1}|)}{|\psi_{\pm1}|} \\ \frac{\partial_\mu(g^{\mu\nu}\partial_\nu|\psi_{\pm2}|)}{|\psi_{\pm2}|}\end{pmatrix} + \frac{\hbar^2}{2}Im\{\sigma^{ab}\omega_{ab\mu}\}g^{\mu\nu}\begin{pmatrix}\frac{\partial_\nu|\psi_{\pm1}|}{|\psi_{\pm1}|} \\ \frac{\partial_\nu|\psi_{\pm2}|}{|\psi_{\pm2}|}\end{pmatrix}\right) \\ +\partial_\mu g^{\mu\nu}\left(\frac{Im\{\sigma^{ab}\omega_{ab\nu}\}}{4}\right) + g^{\mu\nu}Im\{\frac{1}{4}\sigma^{ab}\omega_{ab\mu}\}Im\{\frac{1}{4}\sigma^{ab}\omega_{ab\nu}\}\end{array}\right)$$

$$+Re\left(g^{\mu\beta}g^{\nu\alpha}\left(\begin{array}{c}\frac{ie}{2\hbar}\begin{pmatrix}\alpha_{\pm\beta\alpha1j}\frac{|\psi_{\pm j}|}{|\psi_{\pm1}|} \\ \alpha_{\pm\beta\alpha2j}\frac{|\psi_{\pm j}|}{|\psi_{\pm2}|}\end{pmatrix}F_{\mu\nu} \\ -\frac{i}{4}\alpha_{\pm\beta\alpha}\sigma^{ab}\left(\omega_{ab\mu}\begin{pmatrix}\frac{\partial_\nu|\psi_{\pm1}|}{|\psi_{\pm1}|} \\ \frac{\partial_\nu|\psi_{\pm2}|}{|\psi_{\pm2}|}\end{pmatrix} + \begin{pmatrix}\frac{\partial_\mu\omega_{ab\nu}|\psi_{\pm1}|}{|\psi_{\pm1}|} \\ \frac{\partial_\mu\omega_{ab\nu}|\psi_{\pm2}|}{|\psi_{\pm2}|}\end{pmatrix}\right)\end{array}\right)\right). \tag{13}$$

Afterward, the equation of gravity:

$$R_{\mu\nu} - \frac{1}{2}Rg_{\mu\nu} = \frac{8\pi G}{c^4}\left(Tr\left(\boldsymbol{\tau}_{\mu\nu_{class}} - \boldsymbol{\Lambda}_Q g_{\mu\nu} + \boldsymbol{\Delta\tau}_{\mu\nu_{stress}}\right) + T_{\nu\mu_{em}}\right), \tag{14}$$

is obtained by employing the principle of least action, generalized within the framework of quantum hydrodynamics [11], applied to the motion Equation (12) governing the quantum mass densities.

By following this methodology, the set of equations for gravitationally coupled quantum electrodynamics (G-QED) is accomplished with the following equations [11]:

$$T_{\nu\mu_{em}} = \frac{1}{4\pi}\left(-F_{\mu\lambda}F_\nu{}^\lambda + \frac{1}{4}F_{\lambda\gamma}F^{\lambda\gamma}g_{\mu\nu}\right), \tag{15}$$

$$F_{\mu\nu} = \left( A_{\nu;\mu} - A_{\mu;\nu} \right) = \left( \partial_\mu A_\nu - \partial_\nu A_\mu \right), \tag{16}$$

$$F^{\mu\nu}{}_{;\nu} = -4\pi J^\mu_{em}, \tag{17}$$

$$J^\mu_{em} = -\frac{e\hbar}{im}\overline{\Psi}\gamma^\mu\Psi, \tag{18}$$

$$\left( i\hbar\gamma^\mu \left( \partial_\mu - \frac{i}{4}\sigma^{ab}\omega_{ab\mu} + \frac{ie}{\hbar}A_\mu \right) - mc \right)\Psi = 0, \tag{19}$$

where $T_{\nu\mu_{em}}$ is the electromagnetic tensor and $A_\mu$ is the related vector potential. The tensorial term:

$$Tr\left( \boldsymbol{\tau}_{\mu\nu_{class}} - \boldsymbol{\Lambda}_Q g_{\mu\nu} + \boldsymbol{\Delta\tau}_{\mu\nu_{stress}} \right), \tag{20}$$

in Equation (14) is the energy impulse tensor of the fermion field (Equation (11)) [11], where the terms $-\boldsymbol{\Lambda}_Q g_{\mu\nu} + \boldsymbol{\Delta\tau}_{\mu\nu_{stress}}$, whose detail is provided in [11], contain the quantum contribution encoded into the quantum potential (Equation (13)).

### 2.1. Formal Analysis of Gravity Equation

One of the primary challenges in general relativity is that the energy–momentum tensor density for massive bodies depends on the point-dependent mass density and is limited to the classical expression:

$$T_{\mu\nu} = \frac{mc^2}{\gamma_{(k)}}u_\mu u^\nu, \tag{21}$$

where $(u_\mu = \frac{\gamma}{c}\dot{q}_\mu)$. Consequently, the general form: $T_{\mu\nu} = (p + \varepsilon)u_\mu u_\nu + pg_{\mu\nu}$, remains undefined.

Non-commutative geometry addresses this issue by suggesting that the general form of the energy–momentum tensor density can introduce quantum properties into general relativity, such as the minimum uncertainty. However, the resulting uncertainty relations are not generally equivalent to those provided by quantum mechanics and depend on the definition of $T_{\mu\nu}$. Therefore, a degree of freedom needs to be determined.

To overcome this problem, the Tolman–Oppenheimer–Volkov equation is semiempirical assumed.

In this study, we derived the energy–momentum tensor on the right side of Equation (14) using the quantum hydrodynamic formalism. We accomplished this by imposing the covariance of the quantum mechanical field equations and utilizing the principle of least action. The derived energy–momentum tensor in Equation (14) plays the same role as the Tolman–Oppenheimer–Volkov equation.

The energy–momentum tensor in Equation (20) determines the undefined terms into the energy–momentum tensor of classical general relativity, bringing the quantum physics with it. The correct dilatative action of quantum mechanics is provided through the quantum potential that generates repulsive force against the mass concentration through the term $-\boldsymbol{\Lambda}_Q g_{\mu\nu} + \boldsymbol{\Delta\tau}_{\mu\nu_{stress}}$ in Equation (14). Since the quantum potential action is brought about in the quantum energy–momentum tensor density, consequently, the energy–momentum tensor accurately reproduces the uncertainty relations provided by quantum mechanics.

In principle, the quantum energy tensor density, derived using the quantum hydrodynamic formulation of quantum mechanics, can be considered a specific case of non-commutative geometry. Nevertheless, it worth noting that the formulation of non-commutative geometry is based on semiempirical assumptions, and its constants do not contain the explicit dependence by the fields, but rather require experimental measurements.

On the other hand, the proposed model shows the theoretical connection between the gravitational field of spacetime and the fields residing within it. This feature of the theory offers several advantages, including:

1. Introduction and the possibility to describe the backreaction of fields on gravity from the field dependence of their energy impulse tensor.
2. Description of physical laws at any scale, including scenarios near the Big Bang or within primordial Pre-Big Bang Black Holes.
3. Prediction of small values for the cosmological constant.
4. Self-defined (quintessence-like) cosmological energy pressure density, $\mathbf{\Lambda}_Q$, that emerges from the quantum properties of spacetime.
5. Resolution of the point singularity problem encountered in general relativity [12].

*2.2. Classical and Quantum Spacetime Geometrization*

If we assume that the ST (spacetime) follows the classical equation of motion, without the presence of a quantum potential, we can distribute mass density locally within it, attributing classical characteristics to ST. Conversely, if we assume that the mass distribution in ST is influenced by the quantum potential force, it becomes impossible to freely distribute mass locally, as it becomes coupled to the mass present in the surrounding area, resulting in a quantum mechanical ST. Thus, the properties of ST are determined by the governing law of mass density motion within it.

Since the quantum potential energy contributes to the determination of the curvature of ST, the associated geometry of ST differs from the classical one of the general relativity.

In this regard, it is worth mentioning that since the classical and quantum equations of motion for photons are identical, classical and quantum general relativity coincide in this scenario. This particular characteristic of the electromagnetic field enables the explicit coupling of the photon field to the equation of gravity within the framework of classical general relativity.

For massive bosons and fermions, the complete mathematical coupling between the curvature tensor and the fields can be achieved in the frame of quantum mechanical spacetime geometrization through the covariance condition of the quantum mechanical equations.

When massive particles are present, quantum mechanical spacetime (ST) gives rise to a gravity equation that interacts with the boson and fermion fields of these particles. This gravity equation differs from the classical one due to the inclusion of new effects arising from the presence of the quantum potential.

These effects can encompass several phenomena, such as the absence of a point-like mass density within a black hole, the emergence of a repulsive gravitational force at cosmological distances from a black hole, and the existence of a physically stable vacuum [13]. The presence of cosmological pressure density offers the potential to explain the observed cosmological constant as an indication of the quantum mechanical characteristics of the vacuum, eliminating the need to introduce it as an arbitrary parameter in classical general relativity. Consequently, the cosmological constant lies beyond the classical framework of general relativity, and its existence can be attributed to the quantum mechanical properties of the vacuum.

*2.3. Discussion*

When we transition to curved spacetime, we encounter the following levels of understanding.

Macroscopic classical physics: We encounter the law of evolution of general relativity since general relativity is derived by imposing the covariance of the classical equation of motion (gravitational-inertial mass equivalence) and the least-action condition. Through this approach, we define the gravity of spacetime with classical properties.

In a similar fashion, we define the gravity of spacetime with quantum mechanical properties by leveraging the covariance of the evolution of quantum mechanical fields. This

is equivalent to determining how quantum mechanical mass densities generate gravity within spacetime.

It is important to note that it is not enough to write equations, such as the Klein–Gordon equation, in a generic curved spacetime:

$$\psi^{;\mu}_{;\mu} = (g^{\mu\nu}\partial_\nu\psi)_{;\mu} = \frac{1}{\sqrt{-g}}\partial_\mu\sqrt{-g}(g^{\mu\nu}\partial_\nu\psi) = -\frac{m^2c^2}{\hbar^2}\psi, \tag{22}$$

or to claim that $g^{\mu\nu}$ is defined by classical general relativity in which semiclassical means are introduced. Such an approach is contradictory and does not lead to a self-consistent theory (for instance, the back effect of the field onto the gravity is undefined).

To derive the quantum mechanics in curved spacetime, we must analogously use the covariance of the quantum mechanical equation of evolution and the least-action principle. This allows us to define the gravity of spacetime with quantum mechanical properties (see Table 1).

**Table 1.** The parallelism between classical covariant general relativity and quantum covariant general relativity.

| Classical Equation of Evolution | Covariance of Classical Equation of Evolution | $\Rightarrow$ | Classical General Gravity |
|---|---|---|---|
| (macroscopic scale with $V_{qu} = 0$) | + | $\Rightarrow$ | Covariant classical evolution: spacetime geometry is associated with the dynamics of classical mass densities |
| | Least Action | | $\Uparrow (V_{qu} = 0)$ |
| Quantum Equation of Fields' Evolution | Covariance of Quantum Equation of Evolution | $\Rightarrow$ | $\Uparrow$ |
| | + | | Quantum Mechanical Gravity |
| | Least Action | $\Rightarrow$ | Covariant quantum evolution: spacetime geometry is associated with the dynamics of quantum mass fields |
| Commutation rules for quantum fields' quantization | | | Quantization rules |
| $\Downarrow$ | | | $\Downarrow$ |
| QFT | | | QFT |
| In flat spacetime | | | In curved spacetime |

This procedure leads to quantum mechanics in a spacetime whose curvature is defined by the quantum mechanical fields present within it. In this case, Equation (22) is coupled to the gravity equation, defining a spacetime whose curvature is determined by the field itself. This detail is crucial, as it allows for the description of the backreaction of the fields on gravity, which the semiclassical approximation fails to capture.

The choice between classical spacetime and quantum mechanical spacetime is crucial. The equations governing the evolution of mass densities, which are used to derive the corresponding gravity, are based on either classical mechanics or quantum mechanics covariance. Therefore, if we aim to self-consistently describe quantum mechanics in curved spacetime, we must use covariance conditions applied to the equations of quantum mechanics to define the curvature of quantum mechanical spacetime.

Finally, we can proceed with the second quantization of these fields, treating them as quantum operators based on the field equations of quantum mechanics in curved spacetime.

The main objective of this work was to define the second level of understanding by defining a gravity equation that depends on the quantum mechanical fields present in spacetime.

In conclusion, this paper seeks to establish a theory that aligns quantum mechanics and gravity by considering the intrinsic quantum nature of spacetime. This approach

allows for a consistent treatment of quantum gravity, accounting for the influence of fields on the gravitational field.

### 3. First-Order Gravitationally Coupled QED

The system of G-QED (Equations (14)–(19)) is extremely difficult to handle, but for the specific physical scenario of interest (which involves mass distributions far from the Planckian mass densities), a simplifying perturbative approach can be employed.

The zero-order gravity QGE (Minkowskian spacetime) leads to the ordinary QED.

The first-order G-QED reads [11]:

$$F^{\mu\nu}{}_{;\nu} = -4\pi J^{\mu}_{em}, \tag{23}$$

$$J^{\mu}_{em} = -\frac{e\hbar}{im}\overline{\Psi}\gamma^{\mu}\Psi, \tag{24}$$

$$F_{\mu\nu} = (A_{\nu;\mu} - A_{\mu;\nu}) = (\partial_{\mu}A_{\nu} - \partial_{\nu}A_{\mu}), \tag{25}$$

$$T_{\nu\mu_{em}} = \frac{1}{4\pi}\left(-F_{\mu\lambda}F_{\nu}{}^{\lambda} + \frac{1}{4}F_{\lambda\gamma}F^{\lambda\gamma}g_{\mu\nu}\right), \tag{26}$$

$$\left(i\hbar\gamma^{\mu}\left(\partial_{\mu} + \frac{ie}{\hbar}A_{\mu}\right) - mc\right)\Psi = \frac{\hbar}{8}\gamma^{\alpha}\sigma^{\mu\nu}(\partial_{\alpha}h_{\mu\nu}), \tag{27}$$

where the weak gravity perturbation, $h_{\mu\nu}$, to the Minkowskian metric tensor:

$$g_{\nu\mu} = \eta_{\nu\mu} + h_{\mu\nu} = \begin{bmatrix} 1 & 0 & 0 & 0 \\ 0 & -1 & 0 & 0 \\ 0 & 0 & -1 & 0 \\ 0 & 0 & 0 & -1 \end{bmatrix} + h_{\mu\nu}, \tag{28}$$

is given by the solution of the first-order GE:

$$R^{(1)}_{\mu\nu} - \frac{1}{2}R^{(1)}_{\mu\nu}g_{\mu\nu} = \frac{8\pi G}{c^4}\left(Tr\left(\boldsymbol{\tau}^{(0)}_{\mu\nu_{class}}\right) + T_{\nu\mu_{em}}\right), \tag{29}$$

where the Christoffel symbols reads: $\Gamma^{\beta}_{\mu\alpha} = \frac{1}{2}\eta^{\beta\gamma}(\partial_{\alpha}h_{\gamma\mu} + \partial_{\mu}h_{\gamma\alpha} - \partial_{\gamma}h_{\mu\alpha})$, and where for the *k*-th eigenstate [11]:

$$Tr\left(\boldsymbol{\tau}^{(0)}_{\mu\nu_{class}}\right) = \frac{mc^2}{\gamma}|\psi^{(0)}_{\pm i(k)}|^2 u^{(0)}_{\mu_{\pm i(k)}}\left(u^{(0)}_{\nu_{\pm i(k)}} - \frac{1}{mc}\left(Re\left\{\frac{\hbar}{4}\sigma^{ab}\omega_{ab\nu}\right\} - eA_{\nu}\right)\right), \tag{30}$$

where:

$$u^{(0)}_{\mu_{\pm i(k)}} = \frac{\frac{i\hbar}{2}g_{\mu\alpha}\partial^{\alpha}ln[\frac{\psi^{(0)}_{\pm i(k)}}{\psi^{(0)}_{\pm i(k)}{}^{*}}] + Re\left\{\frac{\hbar}{4}\sigma^{ab}\omega_{ab\mu}\right\} - eA_{\mu}}{mc\sqrt{1 - \frac{V_{qu\pm_{(\psi^{(0)}_{\pm i(k)})}}}{mc^2}}}, \tag{31}$$

where $\psi^{(0)}_{\pm i(k)}$ is the zero-order (flat spacetime) field of the Dirac equation:

$$\left(i\hbar\gamma^{\mu}\left(\partial_{\mu} + \frac{ie}{\hbar}A_{\mu}\right) - mc\right)\Psi^{(0)} = 0 \tag{32}$$

### 4. Discussion: Semi-Quantitative Analysis—Matter–Antimatter Symmetry-Breaking, Quantum Decoherence, Primordial Black Hole Fragmentation, and Mass Expulsion

In the realm of the quantum gravitational equation, the existence of a singularity within a black hole is rendered impossible, making the concept of a point-like mass density

invalid. Consequently, the initial universe system must have possessed a finite initial volume. However, due to the high concentration of mass, it would have given rise to a massive black hole, referred to as the Pre-Big Bang Black Hole (PBBH). The PBBH state was not stationary, and the quantum arrow of time, exemplified by matter–antimatter asymmetry, was, in effect, leading to an irreversible relaxation process that ultimately culminated in the Big Bang.

In the subsequent analysis, we investigate the progression of the Pre-Big Bang Black Hole (PBBH) utilizing the quantum gravitational equation (QGE) coupled with the boson and fermion fields.

*4.1. CPT Inversion and Lepton–Antilepton Symmetry-Breaking in Curved ST*

If we consider the Dirac equation for the fermion field, it is well known that the CPT transformation: $t \rightarrow -t \cup e \rightarrow -e \cup \sigma_i \rightarrow -\sigma_i$, leads to the same Dirac equation for the field of the antiparticle, $\overline{\Psi}$. The formal invariance of the Dirac equation, under CPT inversion, expresses the matter–antimatter symmetry in the Minkowskian spacetime.

If we analyze the gravitational interaction described by the Lagrangian Equation (34), we can see that under the CPT inversion (for which it holds $\Psi \rightarrow \overline{\Psi}$ and $\gamma^\mu \gamma^0 \rightarrow -\gamma^\mu \gamma^0$), the fermion current transforms as:

$$
\begin{aligned}
J^\mu = -\frac{\hbar}{im}\overline{\Psi}\gamma^\mu\Psi = -\frac{\hbar}{im}\Psi^\dagger\gamma^0\gamma^\mu\Psi \rightarrow \\
\rightarrow -\frac{\hbar}{im}\Psi\left(-\gamma^0\gamma^\mu\right)\Psi^\dagger = \frac{\hbar}{im}\Psi\gamma^\mu\overline{\Psi} = -J^\mu
\end{aligned} \tag{33}
$$

while the electromagnetic (EM) current: $eJ^\mu \rightarrow (-e)(-J^\mu) = eJ^\mu$, remains unchanged, so that the gravity interaction Lagrangian under CPT inversion changes as:

$$
\mathcal{L}_I = \frac{\hbar}{8}\sigma^{\mu\nu}J^\kappa\left[\partial_\kappa h_{\mu\nu}\right] \rightarrow -\frac{\hbar}{8}\sigma^{\mu\nu}J^\kappa\left[\partial_\kappa h_{\mu\nu}\right] = -\mathcal{L}_I, \tag{34}
$$

and weakly breaks the matter–antimatter symmetry (the Greek spacetime indices range from 0 to 3). It is worth noting that, at the first order (light particles and Newtonian gravity), the correction to the fermion field due to gravity (in Equation (27)) and the breaking of CPT symmetry is practically null since the antisymmetry of $\sigma^{\mu\nu}$ coupled to the symmetry of the Newtonian metric tensor $h_{\mu\nu}$. Therefore, the difference in mass/energy between a fermion and its antiparticle becomes significant only for very heavy fermions, which are sources of strong gravitational fields (such as those present in PBBH) with masses of the order of the Planck mass or even bigger.

Furthermore, since the magnetic moment of leptons is dependent on their masses, it follows that the slight difference in the magnetic moment of a lepton and its corresponding antiparticle becomes increasingly larger as the mass of the lepton increases. This provides a potential means of testing the theory.

When considering only the dynamics of electro-gravity, in which only gravity and electromagnetic forces are considered, the presence of the electro-gravity coupling term in Equation (34) requires us to consider the theoretical possibility that a graviton (which is very light) may be released/absorbed in the process of lepton–antilepton annihilation, as well as in the inverse process. This is necessary in order to account for the difference in gravitational mass between fermions and antifermions.

The matter–antimatter (fermion–antifermion) asymmetry induced by gravity allows the formulation of a mechanism for the realization of baryonic asymmetry in the early universe. Generally speaking, the gravitational generation of baryonic asymmetry in the high-energy states of PBBH is produced by the gravitational interaction with all the fields of bosons and fermions.

The physical imbalance between matter and antimatter in the quantum PBBH immediately leads to the breaking of time inversion symmetry in the QGE–fermion–boson fields system of equations. Consequently, the highly ordered quantum configuration of PBBH undergoes a time-directional, irreversible evolution toward a less ordered conformation,

leading to the appearance of the quantum arrow of time and the irreversible randomization of energy that generated the Big Bang. The production of many residual light particles, which constitute the mass difference in the annihilation of high-energy matter–antimatter states, makes the inverse process highly unlikely, requiring the simultaneous grouping of many product particles.

*4.2. Cosmological Constant, Fermion–Antifermion Annihilation, and Matter–Antimatter Asymmetry*

As illustrated by Equation (20), particles possessing rest mass exhibit a non-zero cosmological pressure density (CPD), $\mathbf{\Lambda}_Q$. Consequently, when a fermion and its corresponding antiparticle annihilate, the gravitational field undergoes a transition from one characterized by a non-zero CPD to one featuring a zero CPD, since the emitted photons possess a zero CTD. The absence of a CPD for photons can be attributed to the fact that the QGE governing the electromagnetic photon field simplifies to the classical expression found in general relativity.

Conversely, when a photon generates an electron–positron pair, the gravitational field associated with the resulting massive particle and antiparticle is formed. If the photon possesses a null scalar Ricci curvature, R [14], and CPD, then the scalar curvature of the electron and positron's gravitational field is non-zero, along with a non-zero CPD.

The formation of an electron–positron pair and the subsequent alteration of the gravitational field lead to an energy change, attributing a gravitational contribution to the masses of both the particle and the antiparticle. Hence, during their annihilation, it is anticipated that a low-energy graviton will be emitted.

Furthermore, considering that the energy (as indicated in Equation (34)) has opposite signs for particles and antiparticles, there exists a slight disparity in their masses.

For light fermions, the disparity in mass mentioned earlier is insignificant since they serve as sources for the Newtonian gravity field. As we approach the Minkowskian limit, the symmetry between matter and antimatter becomes asymptotically established. Consequently, the discrepancy in mass between particles and antiparticles diminishes progressively as we transition from heavier to lighter particles within each particle family.

This pattern of behavior implies that the asymmetry between matter and antimatter could have been notable in the vacuum states of extremely high energy prior to the Big Bang. Within the pre-Big Bang horizon, the high-energy fermion state, surpassing the Planck mass, comprised black holes formed by fermions and antifermions. Through their annihilation, these black holes emitted a burst of lighter fermions that accounted for the disparity in mass between them.

If we consider, by assuming a contrary position, that the matter–antimatter symmetry was preserved within the PBBH, it implies that the subsequent universe following the Big Bang would possess a cosmological constant of zero. This is because in the absence of massive particles (given that photons have a cosmological pressure tensor density of zero), there would be no contribution to the cosmological constant and the vacuum state would have collapsed into the polymer branched phase [15].

Nevertheless, according to the quantum gravity equation, the existence of a non-zero cosmological constant, which we observe in the present-day universe, serves as evidence for the matter–antimatter asymmetry within the initial PBBH and the quantum properties of spacetime gravity. In this context, the cosmological pressure density represents a quantum contribution to the gravitational field.

*4.3. High-Temperature Quantum Decoherence, PBBH Fragmentation, and Mass Expulsion*

The annihilation of very heavy PBBH massive vacuum states (consisting of high-energy fermions and massive bosons), with their respective non-symmetric antimatter states during the Big Bang, should have resulted in:

1. The emission of gravitational boson waves, which contribute to the content of dark energy in the present universe.
2. The production of fragments of SMBHs from the initial PBBH.

3. The release of residual low-energy fermions, which constitute the baryonic and dark matter of the present universe.

All of these leftover parts can be considered as the "ashes of the Big Bang".

Since the matter–antimatter annihilation is irreversible (meaning the backward process of fermion–antifermion formation is not as likely as annihilation due to the very large number of products), the energy, associated with the products of the annihilation, undergoes randomization. This results in an increase of the fluctuation amplitude up to a temperature of:

$$m_{PBBH} \kappa R_g \sqrt{\frac{kT}{2\hbar}} \gg \left| \frac{\partial V_{qu(n)}}{\partial q_i} \right|, \tag{35}$$

at which the quantum potential force, $\frac{\partial V_{qu(n)}}{\partial q_i}$, is deeply perturbed and the quantum coherence breaks down [13] ($R_g$ and $m_{PBBH}$ are the gravitational radius of the PBBH and its mass, respectively, $k$ is the Boltzmann constant, $T$ is the temperature, $\hbar$ is the Planck constant, and $\kappa$ is the inverse of the quantum friction coefficient [13]).

Thus, as the PBBH experiences quantum decoherence and transitions into a classical entity, it breaks apart and releases supermassive black hole (SMBH) snippets with surplus matter around them.

The fragmentation of the PBBH and the ejection of mass into the expanded spacetime with low curvature outside the gravitational radius of each SMBH are two interconnected stages of a unified process. As the temperature of the PBBH increases due to matter–antimatter annihilation, the expanding force of the quantum potential, $V_{qu}$:

$$-\partial_\mu V_{qu} \sim \frac{\hbar^2}{m} \partial_\mu \frac{\partial_\nu \partial^\nu |\psi|}{|\psi|}, \tag{36}$$

supported by the thermal expansive force due to the thermodynamic potential, $V_{therm}$:

$$-\partial_\mu V_{therm} \sim -D \frac{1}{|\psi|} \partial_\mu |\psi|, \tag{37}$$

produces the mass expansion (with the decrease of its mean density inside the sphere of the PBBH gravitational radius) far below the critical value:

$$\frac{3 m_{PBBH}}{4\pi R_g^3} = \frac{3 c^6}{32 \pi G^3 m_{PBBH}^2}, \tag{38}$$

producing expulsion of mass in the form of a SMBH and in the low-gravity space between the fragments of the SMBH.

As each fragment of a SMBH possesses a lower gravitational pull compared to the original PBBH, the mass within each fragment continues to disperse further. Throughout this progression, the gravitational radius of each SMBH diminishes, resulting in more mass being located outside of it. This external mass then undergoes rotational motion within the gravitational well of the SMBH.

The process of SMBHs contracting in size, accompanied by mass expulsion, concludes when a new state of equilibrium is achieved between the gravitational force and the repulsive quantum force. This equilibrium is attained due to the decrease in temperature caused by the expulsion of matter.

This process gives rise to SMBHs harboring external mass that orbits within their gravitational wells, thereby providing an explanation for the formation of observed galaxies. Additionally, it elucidates how SMBHs represent a prevalent cosmological configuration and how they could form without the need for mass accretion.

This model is consistent with recent findings from the James Webb Space Telescope [16,17], which detected galaxies at far redshifts hosting SMBHs at their centers shortly after the Big Bang, within the first billion years. The observation extends progressively from just under

one billion years to 0.7–0.6 billion years. There were challenges faced by astrophysics in explaining the formation and rapid growth of these SMBHs during that early epoch, as well as attempts to identify specific phenomena accelerating their mass increase [18,19]. To attain BH masses $<\sim 10^9 M_\odot$ within the typical star formation duration of $<\sim 1$ Gyr of the host galaxy without the dynamical friction process is challenging, especially at high redshifts of $z > 6$ or for over-massive BHs that are upper outliers of the average Magorrian relationship. In such a case, the BH growth must proceed at appreciably high Eddington ratios of $\lambda >\sim 1$ or starting from heavy BH seeds of $M_\bullet \sim 10^{3-5} M_\odot$. This instance can be partially justified theoretically but struggles somewhat against the present observational estimates. This defies expectations of the mass accretion mechanism, which would need a decrease in mass content as we approach the moment of the Big Bang.

As we trace our steps backward in time toward this pivotal moment, astrophysicists may encounter the compelling need to reconsider the prevailing mass accretion hypothesis. Embracing this evolving perspective could benefit from exploring new theoretical frameworks that might offer valuable insights and enrich our understanding of SMBH formation.

## 5. Discussion

According to reference [13], non-commutative gravity emerges from the covariance of quantum mechanics, incorporating uncertainty relations and the maximum speed of light and information transmission. This concept aligns with the stochastic quantum hydrodynamic model (SQHM) [20], elucidating how the gravitational background induces quantum decoherence and possibly gives rise to a macroscopic "coarse-grained" classical reality.

Within the framework of this unified approach, where the maximum attainable velocity cannot exceed the speed of light, such as

$$\dot{x} \leq c, \tag{39}$$

coupled with the uncertainty relations that necessitate

$$\dot{x} \geq \Delta \dot{x} = \frac{\Delta p}{m} = \frac{\hbar}{2m\Delta x}, \tag{40}$$

Leads to $\frac{\hbar}{2m\Delta x} \leq c$ and, consequently, to:

$$\Delta x > \frac{\hbar}{2mc} = \frac{R_c}{2}, \tag{41}$$

where $R_c$ is the Compton length.

Identity Equation (41) shows that the minimum dimension of a body is half of its Compton length. Transferring this output to BH we find that, since, in order to form a BH, all the mass must be inside the gravitational radius, $R_g$, we must have that:

$$R_g = \frac{2Gm}{c^2} > \frac{\Delta x}{2} = r_{\min} = \frac{R_c}{4}, \tag{42}$$

and thus, that:

$$\frac{R_c}{4R_g} = \frac{\hbar}{8mcR_g} = \frac{\hbar c}{8m^2 G} = \pi \frac{m_p^2}{m^2} < 1 \tag{43}$$

leading to the condition for the black hole mass, $m_p$, is the minimum one possible for BHs at $T = 0$.

For temperatures greater than zero, the Planck mass black hole becomes unstable. This instability arises as the expansive thermal force combines with the quantum force, pushing a portion of the mass beyond the gravitational radius. Consequently, the residual

mass inside is unable to generate sufficient gravitational potential to form the black hole, ultimately resulting in its evaporation.

The validity of the result in Equation (41) is supported by quantum mechanical gravity [13], which demonstrates that when mass density is compressed into a sphere with a diameter equal to the Compton length, it generates a quantum potential force that precisely counteracts the compressive gravitational force. As the BH of Planck mass represents the lightest configuration, with its mass compressed within a sphere of half the Compton wavelength, it follows that black holes of higher mass exhibit their mass compressed into a sphere of smaller diameter. Consequently, the theory raises concerns regarding the consideration of the Planck length as the smallest discrete elemental volume of spacetime.

This output holds significant importance, as it forms the basis for both loop quantum gravity and non-commutative string theories. These theories rely on the theory that there exists an absolute limitation on length measurements in quantum gravity. While, in principle, it is correct to steer clear of theoretically unrealizable infinite and infinitesimal concepts, the fundamental arguments of these theories assume that to pinpoint a particle within a sphere of a Planck length radius, an energy greater than the Planck mass is required that shields whatever occurs within the Schwarzschild radius. Consequently, this represents the smallest 'quanta' of space and time. The critical aspect is that a black hole of Planck mass stands as the lightest, and any existing black hole compresses its mass into a nucleus smaller than the Planck length [13].

Since, from a discretization standpoint, it is not possible to compress anything in a volume smaller than the elemental one, this makes it impossible to compress the mass of large black holes within a sphere of half the Compton diameter, consequently preventing the achievement of gravitational equilibrium.

This compression is only feasible if spacetime discretization allows elemental cells of smaller volume. In my opinion, the solution lies in avoiding the confusion between the minimum measurable distance and the minimum texture length of spacetime.

Finally, it is worth noting that the current theory leads to the assumption that the minimum texture length of spacetime corresponds to the Compton length of the maximum possible mass, which is the mass of the universe. Consequently, we have a criterion to rationalize the mass of the universe—why it is not higher than its value—and it is intricately linked to the minimum length of the discrete spacetime texture.

## 6. Conclusions

The quantum spacetime geometrization has the capability to give rise to a gravity equation that is analytically coupled to the fermion and boson fields. This achievement is made possible by incorporating two fundamental principles in the process of generalization: field equation covariance and the least-action principle.

The theory establishes the reciprocal influence of fields on the gravitational equation, thereby defining the impact of fields on gravity. The gravitational backreaction of the fields is determined through the energy tensor density of the fields, resulting in a non-commutative model. The coupled system of gravity–field equations does not rely on semiclassical approximations or weak gravity conditions. The backreaction of the fields is accounted for at any level of approximation, enabling the description of gravity and physical laws across all distance scales and under conditions of high gravity, including the Big Bang scenario. On a cosmological scale, the model resolves the point singularity issue associated with black holes. At the scale of elementary particles, the quantization of the field variables gives rise to an operational system of gravity–field equations capable of describing high-energy excited states of the vacuum, leading to significant spacetime curvature. The weak gravity limit enables the calculation of gravitational corrections to QED and, potentially, to the standard model as well.

Furthermore, it offers an explanation for the presence of the quintessence-like cosmological pressure density and the breaking of matter–antimatter symmetry at high energies.

In the cosmological model, the theory provides an explanation for the formation of supermassive black holes, surrounded by their own galaxy, directly from the Big Bang dynamics without the need for mass accretion. This model also aligns with the recent observations made by the James Webb Space Telescope, which provide support for the early formation of galactic configurations shortly after the Big Bang.

It is noteworthy to observe that, similarly to classical general relativity, which couples with the electromagnetic field and is formally unified in the context of five-dimensional gravity, as proposed by Theodor Kaluza [21], the system of equations for the Quantum Gravity Extension of the Standard Model establishes the mathematical groundwork for developing a unique, multi-dimensional gravity-like equation, which encompasses the description of fundamental interactions and particle fields.

**Funding:** This research received no external funding.

**Data Availability Statement:** No new data were created or analyzed in this study.

**Conflicts of Interest:** The authors declare no conflict of interest.

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
