# Peer review of "Quantum Spacetime Geometrization: QED at High Curvature and Direct Formation of Supermassive Black Holes from the Big Bang"

_quantumrep, doi:10.3390/quantum6010002_

Round 1

Reviewer 1 Report

Comments and Suggestions for Authors

In the submitted manuscript the author is making an attempt to describe the gravitational interaction within quantum theory via geometrization of space-time. The quantum theory here is QED, although in principle the approach may be extended to the SM of particle physics.

The gravitational backreaction of matter fields is properly taken into account without relying on weak gravity conditions nor on semiclassical approximations. Formation of supermassive BHs is briefly discussed as well.

I find the work quite interesting, although a few points must be improved before I recommend the draft for publication. Those are the following:

1. I feel the bibliography is rather poor. A more extensive list of references should be provided by the author.

2. There are too many self-citations as 8 out of 23 Refs. (almost 1/3) belong to the author. If I am not mistaken, according to the policy of the MPDI journals, only up to 10 % of the total number is allowed.

3. The Refs. 21 and 22 on the James Webb Space Telescope seem to be missing. Moreover, there are no figures showing how the model is in agreement with the data, as claimed in the text. 

4. In the Conclusions section it is written that the approach results in a non-commutative model. How is that model different in comparison to the non-commutative field theories obtained within superstring theory? Perhaps a brief comment may be added here.

Therefore, I am asking for minor revisions.

Author Response

The answer to the reviewer is in the file: Ref 1 with reply.docx.

do not consider the second one "author-coverletter-32861709.v1.docx" there just for mistake.

Reviewer 2 Report

Comments and Suggestions for Authors

The paper deals with an approach to quantum gravity, called "quantum space geometrization", and in particular discusses its application to  the cosmological constant problem and the matter-antimatter asymmetry in the universe.  

The equations for fermionic matter and gravity approach are very briefly described in section 2, from which it looks like they are simply a rewriting of the standard Dirac equation on curved background and Einstein equation in a hydronamics-inspired parametrization. Some parts of the equations are not even defined, just referring to some of the author' s previous publications. This is quite confusing because I was not able to understand why the equations should not be regarded as completely standard textbook material and give some novel insight about the aforementioned problems. 

The rest of the paper is a redundunt discussion where the author tries to support the validity of his method and deals with the mentioned applications. However, he provides no rigorous mathematical treatment, limiting himself to rather questionable ponderings. In particular, I was impressed by the claim that the standard fermion coupling to gravity breaks CPT invariance, which is a wrong one in my opinion. I think equation (4.2), apart from lacking correct contraction of spinor indices, can be easily proven wrong by taking into account the correct transformations for C, P and T.

Finally, a lot of sentences present such a poor formulation as to sometimes make it really hard to understand what the author is arguing.

For these reasons, I think the author should reconsider both the form and the content of his draft.

Comments on the Quality of English Language

 A lot of sentences present such a poor formulation as to sometimes make it really hard to understand what the author is arguing.

Author Response

the answers to the reviewer are in the attached file:Ref 2 with reply.docx

Do not consider the second file "author-coverletter-32861743.v1.docx" there just for mistake.

Reviewer 3 Report

Comments and Suggestions for Authors

This manuscript, while considers interesting issues for a real research, has not enough soundness for publication. Several fundamental issues are argued to be addressed but via some not so sound conceptual and mathematical argumentation. The claimed addresses to the issues such as matter-antimatter asymmetry, the origin of the cosmological constant, formation of supermassive black holes and so on, are not acceptable in the absence of some arguments and justifications in confrontation with observational data and also experiments. For instance, after realization of the Event Horizon Telescope data for M87* and Sgr A* supermassive black holes (shadow cast of these objects), it is natural to ask whether the speculations and claims in this manuscript have anything to do with the real world. For these reason, I think the present manuscript deals with some fundamental issues without a conceptually and mathematically sound reasoning and justification, where there is no confrontation of the speculations with observations/experiments in accordance with the real world.  So, in my opinion this manuscript is not suitable for publication in Quantum Reports and should be rejected.  

Author Response

the answers to the reviewer are in the attached file: Ref 3 with reply.docx

Do not consider the second file "author-coverletter-32954109.v1.docx" there just for mistake.

Reviewer 4 Report

Comments and Suggestions for Authors

Manuscript ID: quantumrep-2674117

Dear Editor,

I have completed the review of the manuscript titled "Quantum Spacetime Geometrization: QED at high curvature and Supermassive black holes in the Early Universe." The article presents an innovative approach that utilizes quantum hydrodynamic formalism to achieve the geometrization of spacetime, addressing key issues in the intersection of quantum theory and general relativity.

The article outlines the methodology employed by the author to connect quantum field equations and spacetime geometry. The integration of fundamental principles, namely the covariance of quantum field equations and the principle of least action, results in a theoretical framework that allows for the calculation of gravitational corrections to Quantum Electrodynamics (QED). The manuscript also explores the implications of the theory, offering explanations for the cosmological constant, matter-antimatter asymmetry, and the formation of supermassive black holes. Furthermore, as discussed in the introduction, the study highlights the challenges faced in unifying general relativity and quantum theory. It provides a critical review of existing approaches such as string theory and loop quantum gravity, identifying their limitations. This sets a clear context for the author's novel approach and the unique contributions made by the proposed theory. The manuscript further addresses the shortcomings of loop quantum gravity, string theory, and quantum field theory in curved spacetime, providing a well-rounded assessment of the current state of theoretical physics. This critical evaluation enhances the significance of the proposed quantum hydrodynamic approach and its potential to address existing challenges.

The detailed analysis of the problems associated with general relativity in dealing with the quantum nature of matter is insightful. The author convincingly argues for the necessity of a more comprehensive theory that can accommodate quantum effects in the gravitational field, addressing the limitations of the classical expressions within general relativity. The recent studies referenced by the author [10,11] provide a foundation for the proposed theory, showcasing its ability to describe the spacetime geometry through a gravity equation that incorporates quantum mechanics. The incorporation of non-commutative terms into the energy-momentum tensor, resulting in a self-defined cosmological energy pressure density, is a notable contribution. The advantages of the proposed theory, including the resolution of point singularities, the explanation of repulsive Newtonian gravity at large distances, and the potential advancements in Quantum Electrodynamics, are well articulated. The analytical relationship between gravity and fields, enabling the description of field evolution across different physical scales, adds depth to the proposed model.

In summary, the manuscript is well-written, logically structured, and presents a compelling argument for the novel quantum hydrodynamic approach. The author effectively addresses the limitations of existing theories and provides a clear and innovative path forward. I recommend the acceptance of this manuscript for publication, given its potential to significantly contribute to the ongoing discourse in theoretical physics.

Thank you for considering my review.

Sincerely.

Author Response

The answers to the reviewer are in the attached file.

Round 2

Reviewer 3 Report

Comments and Suggestions for Authors

The quality of the manuscript has been boosted considerably by revision based on my previous comments. The newly added materials along with the extended calculations in the  "complete text" have convinced me to accept this revised manuscript for publication in Quantum Reports.  

Author Response

The referee's response indicates a thorough and accurate understanding of the significance of the work. There are no objections raised.